# Politics overwhelms science in the Covid-19 pandemic: Evidence from the whole coverage of the Italian quality newspapers

Stefano Crabu[1]*, Paolo Giardullo[2], Andrea Sciandra[3], Federico Neresini[2]

**1** Department of Design, Politecnico di Milano, Milano, Italy, **2** Department of Philosophy, Sociology, Education and Applied Psychology, University of Padova, Padova, Italy, **3** Department of Communication and Economics, University of Modena and Reggio Emilia, Reggio Emilia, Italy

* stefano.crabu@polimi.it

**Data Availability Statement:** We made the "Sars-Cov-2 general corpus" data publicly available. We provided all information necessary for interested researchers to gain access to the data: date, source

## Abstract

The SARS-CoV-2 pandemic has emerged as one of the most dramatic health crises of recent decades. This paper treats mainstream news about the current pandemic as a valuable entry point for analyzing the relationship between science and politics in the public sphere, where the outbreak must be both understood and confronted through appropriate public-health policy decisions. In doing so, the paper aims to examine which actors, institutions, and experts dominate the SARS-CoV-2 media narratives, with particular attention to the roles of political, medical, and scientific actors and institutions within the pandemic crisis. The study relies on a large dataset consisting of all SARS-CoV-2 articles published by eight major Italian national newspapers between January 1, 2020 and June 15, 2020. These articles underwent a quantitative analysis based on a topic modeling technique. The topic modeling outputs were further analyzed by innovatively combining ad-hoc metrics and a classifier based on the stacking ensemble method (combining regularized logistic regression and linear stochastic gradient descent) for quantifying scientific salience. This enabled the identification of relevant topics and the analysis of the roles that different actors and institutions engaged in making sense of the pandemic. The results show how the health emergency has been addressed primarily in terms of political regulation and concerns and only marginally as a scientific matter. Hence, science has been overwhelmed by politics, which, in media narratives, exerts a moral as well as regulatory authority. Media narratives exclude neither scientific issues nor scientific experts; rather, they configure them as a subsidiary body of knowledge and expertise to be mobilized as an ancillary, impersonal institution useful for legitimizing the expansion of political jurisdiction over the governance of the emergency.

## Introduction

The ongoing Covid-19 pandemic has been recognized as one of the most dramatic global health crises of the last decades. Beyond its social and economic impacts, the pandemic is

(newspaper), and URL of each of the 58,646 articles belonging to the "SARS-CoV-2 general corpus." This dataset is adequate to replicate our study findings, and it has been deposited in a public repository, Zenodo, under the following doi: 10.5281/zenodo.4624096 (Data from: Politics overwhelms science in the Covid-19 pandemic: evidence from the whole coverage of the Italian quality newspapers).

**Funding:** The authors received no specific funding for this work.

**Competing interests:** The authors have declared that no competing interests exist.

redefining the relationships between science, public policy, and society, the full extent and consequences of which remain to be seen [1–6]. Although the worldwide plea to "follow the scientists' advice" immediately resonated in media spaces and public debate less than one week after the first Covid-19 case was diagnosed, it is difficult to overlook the uncertainties that have accompanied scientific advising to governmental decision-making. In this scenario, scholars from various fields have noticed that the general public are confused and over-whelmed by a breadth of public narrative that blurs the boundaries between scientific, medical, political, and economic discourses [7–10]. Indeed, narratives in diverse media sources seem to have played a critical role in shaping the collective meaning of this emerging global health emergency. As Rosenberg [11] pointed out in the pioneering work tellingly titled "What is an Epidemic?", epidemics and pandemics are strictly related to public health and narratives about medicine and life sciences depicting disease histories and patients' clinical trajectories. Accordingly, in everyday (current) pandemic life, people are exposed to a plurality of data and interpretations through which mainstream media promote diverse implicit or explicit inter-pretative frames about the health emergency.

In this perspective, newspaper articles represent an important resource for analyzing how societies understand the origin of an outbreak; the conditions under which concerned groups of people, as well as scientific and medical institutions, enter into public decision-making pro-cesses during infectious disease outbreaks; and the ways in which citizens' responsibility in fac-ing pandemics is shaped as collective endeavor. Mainstream media narratives are not only influenced by diverse social actors and stakeholders (e.g., policymakers, political opinion lead-ers, researchers, patients' families and organisations, and various interest groups) as "claim-makers" about their own perspectives in addressing health crisis—they also contribute to ori-ent the agenda of public discourses and reinforce or contrast what is going on within con-cerned scientific domains. In this regard, contemporary media are generative elements engaged in the exchange, reproduction and transformation of the (social) meaning of health-, medicine- and pandemic-related content [12–14]. This is also clearly demonstrated by a large research body in social sciences about the last (potential) pandemics [15–18]. Within this research stream, scholars showed how media narratives about pandemic and health crisis fol-low a recurrent and peculiar pattern, or cycle: it goes from the declaration of the alarm to a more reassuring register, independently by the tendency of the media coverage and the specific threats [15–18]. A fast sequence of "scary news" (e.g. the growing number of contagions, death tolls and description of hard clinical consequence for people infected) is followed by a series of narratives intended to relieve the audience from anxiety, assuring that health authorities have the right tools to contrast the emergency and to contain the contagion.

Relying on this body of research, the present article contributes to the ongoing debate over the decision-making and public shaping of science policy related to the Covid-19 pandemic, referring to Italian mainstream newspapers as a valuable source of empirical data. It assumes that pandemics are strictly interconnected with the broader media, cultural, and political land-scapes [15–18]. Thus it offers an empirically-based study able to contribute to the current debate about the so-called "infodemic" [19–21] centered on how the massive production of information in digital and physical media environments can affect the public meanings, per-ception and governance of a disease outbreak. Indeed, public narratives can shape, reproduce and reinforce what seems possible (knowledge) and desirable (imaginaries) as well as what seems appropriate or inappropriate (norms, values and beliefs). Hence, under this perspective what is at stake is not so much to describe in which ways the media depict science and pan-demic public policies. Rather, although it is well recognized that the timing of news making is quite different from that of scientific knowledge-making [20], what is relevant here is that on the one hand, the media can be observed as active agents contributing to the management of

the pandemic (i.e. the media as performative agent; see [22, 23]); on the other hand, they represent a source of data for studying those processes. Therefore, the media are here understood both as discursive arenas engaged in co-shaping public responses to the pandemic and as a data source for analyzing how political institutions manage relations with technical and scientific regulatory bodies and scientific communities for the sake of public health [24–26]. This is particularly urgent when there is a relative lack of curative and preventive treatments to face the Covid-19 disease—as occurred especially during the so-called "first wave"—when policies and protocols against the spreading of the virus were primarily rooted in lifestyle and behavioral changes, that is social norms and convention, whose plausibility and legitimacy are widely debated by mainstream media. Accordingly, it is crucial to analyze the Covid-19 media accounts circulating in the Italian public sphere during the first wave of the Covid-19 pandemic (January-June 2020, see "Datasets Section"), which constitute a fruitful empirical time frame for understanding the re-articulation of the entanglement between science and politics, due to the high scientific uncertainty, together with a well-recognized relative paucity of evidence-based lessons to treat Covid-19 patients [1].

The analysis has been realized by developing an extensive machine learning technique-based study about the SARS-CoV-2 coverage of eight major national Italian newspapers. Firstly, it has been applied first topic modelling techniques based on Latent Dirichlet Allocation (LDA [27–29], see "Datasets" and "Data Analysis" Sections), and then a supervised machine learning classifier (see Section "Data Analysis").

The study addresses the following research question:

*RQ1: Which kind of domains are primarily mobilized by mainstream media in addressing SARS-CoV-2 related issues? Or more specifically, which domain, between the scientific and the political one, is prevalent in media discourses over Covid-19 pandemic?*

It is worth mentioning that this research question raises the problem of defining the theoretical and empirical criteria necessary to distinguish between the domain of science and that of politics [30]. For the purpose of this study, this issue cannot be neglected, but it has been addressed pragmatically by taking into account two main aspects. Firstly, the two domains (i.e. science and politics) are certainly not clearly separable within the media public discourse. On the contrary, it is a question of degree of entanglement: that is, science and politics are always mixed not only with each other but also together with many other thematic areas, such as economics or sport. This means that even if an article mainly addresses scientific issues and is thus classified as featuring predominantly scientific content, it may also include references to politics or economics or other topics. The second aspect to be considered is that this study relies on a large corpus of articles (i.e. 58,646 articles, see "Datasets" section), and therefore it requires to rely deeply on automatic techniques. It should be noted that using machine learning techniques allows researchers to combine these two aspects in a relatively consistent and simple way. In particular, topic modeling allows the analyst to both identify fairly quickly the presence of a certain number of thematic areas–i.e., the topics–within a corpus and to assess which topic is the most represented one within a given article. Topics can be then assigned to specific thematic domains, such as science or politics, through the interpretation of researchers who assign each article to the topic that is most represented within its text. At this point, it will be sufficient to assign the article to the thematic domain that includes the various related topics in order to have as many article collections (corpora) as are the domains being analyzed. Sets of corpora for each thematic domain are thus made available for further investigation (more details are provided in the "Datasets" Section). Of course, this does not theoretically solve the so-called "demarcation problem", i.e. the identification of the features that clearly distinguish

science from other socio-cultural domains; however, assuming that media discourses are the focus and the source of this analysis, it is suitable to see demarcation as a matter of degrees of intersection, rather than of clear distinctions. Indeed, the specificity of news coverage about science relies on the fact that actors operating in the media arena are interested in making scientific content relevant for society at large [31–33]. For this reason, there is no obligation to solve the demarcation problem in philosophical or, generally speaking, theoretical terms; rather, it is necessary to find some operational criteria able to quantify the extent to which scientific content is present in a given newspaper article. In other words, deciding *a priori* what exactly science is not relevant to the purposes of this study; on the contrary, it is possible to avoid ontological questions and consider what is taken as 'scientific' by common readers, that is, what is presented and/or represented as 'science' by the media.

Following this approach in mapping the dominant thematic domains within media discourses as outlined in RQ1, the paper also aims to determine who is in charge of reassuring and offering cognitive and evidential resources to the public, so that people can deal with everyday life during the pandemic. This aim corresponds to the second research question:

> *RQ2: What types of actors, institutions, and expertise dominate the SARS-CoV-2 media narratives?*

In addressing RQs, the Italian case is particularly relevant for three main substantive reasons: (a) Italy was one of the first European countries to experience widespread SARS-CoV-2 outbreaks [34, 35]; (b) beside China and South Korea, Italy was the first to adopt drastic measures to contain the contagion [34, 35]; and (c) leading international Governments (primarily France, Spain, and Germany), the WHO and other NGOs and health organizations have recognized Italy as cutting-edge in its implementation of effective best practices for pandemic management [34, 35]. Overall, these three dimensions configure Italy as an emblematic case study to grasp how science and politics interact within the media sphere, and the ways in which public decision-making and collective meanings about the health-related crisis are thus configured.

The paper is organized as follows: the next section describes the research design based on topic modelling technique [27, 28] to measure both the relevance of scientific content within media discourses, and to compare political and scientific contributions to shaping those discourses. After discussing the methodological framework, the paper delves in the analysis of thematic domains as understood through topic modelling, further emphasizing the role of different actors engaged in making sense of the pandemic and outlining the management of the subsequent crises through ad hoc metrics applied directly to the corpus under study. The closing section, by synthesizing the most significant findings, discusses the relevance of the methodological and analytical perspectives that have been exploited to understand the pandemic as a multi-layered phenomenon where news-making, political decision-making, and scientific endeavors are thoroughly interlaced.

## Materials and methods

This paper provides an understanding of the mainstream media narratives as a constitutive environment to make sense of the pandemic and co-define the differential distribution of public jurisdiction between scientists and politicians to manage the SARS-CoV-2 health crisis [36]. A research design exploiting a massive corpus composed of articles from eight major Italian newspapers is used to answer the RQs. As a whole, this group of newspapers well represents the mainstream media's array of political and cultural positions within Italian society:

progressive (i.e., La Repubblica), moderate (i.e., Il Corriere della Sera, La Stampa, Il Mattino di Napoli, Il Messaggero), conservative (i.e., Il Giornale), neoliberal (i.e., Il Sole 24 Ore), and Catholic (i.e., L 'Avvenire) [see 37 and S1 Table in S1 File].

In order to answer the RQs, the analysis strategy involves the use of different corpora with increasing levels of specificity with respect to the issues addressed (see "Datasets Section"). The corpora were analyzed using machine learning techniques to obtain a classification according to their content (scientific or non-scientific) and to identify the topics covered by the articles. The topic modeling has also allowed to focus on articles belonging to thematic domains of interest and to measure the presence of specific personalities and institutions within them.

The methodological novelty of this work relies precisely in the combination of machine learning techniques and their outputs (such as posterior word-topic probabilities) to extract the main actors of the Italian pandemic and measure their relevance through the frequency of their occurrence in newspapers.

## Datasets

The analysis described below consists of three nested corpora of newspaper articles: i) a corpus containing all the articles published in the timespan between January 1, 2020 and June 15, 2020 (hereafter "total corpus", see Section 3.1); ii) the corpus of articles selected within the "total corpus" according to the occurrence of keywords related to SARS-CoV-2 (hereafter "SARS-CoV-2 general corpus") and; iii) the "SARS-CoV-2 focused corpus" derived from the previous one through the selection of articles related to the "politics", "science", and "medicine" thematic domains emerging from the topic modelling (see Fig 1). More specifically, the

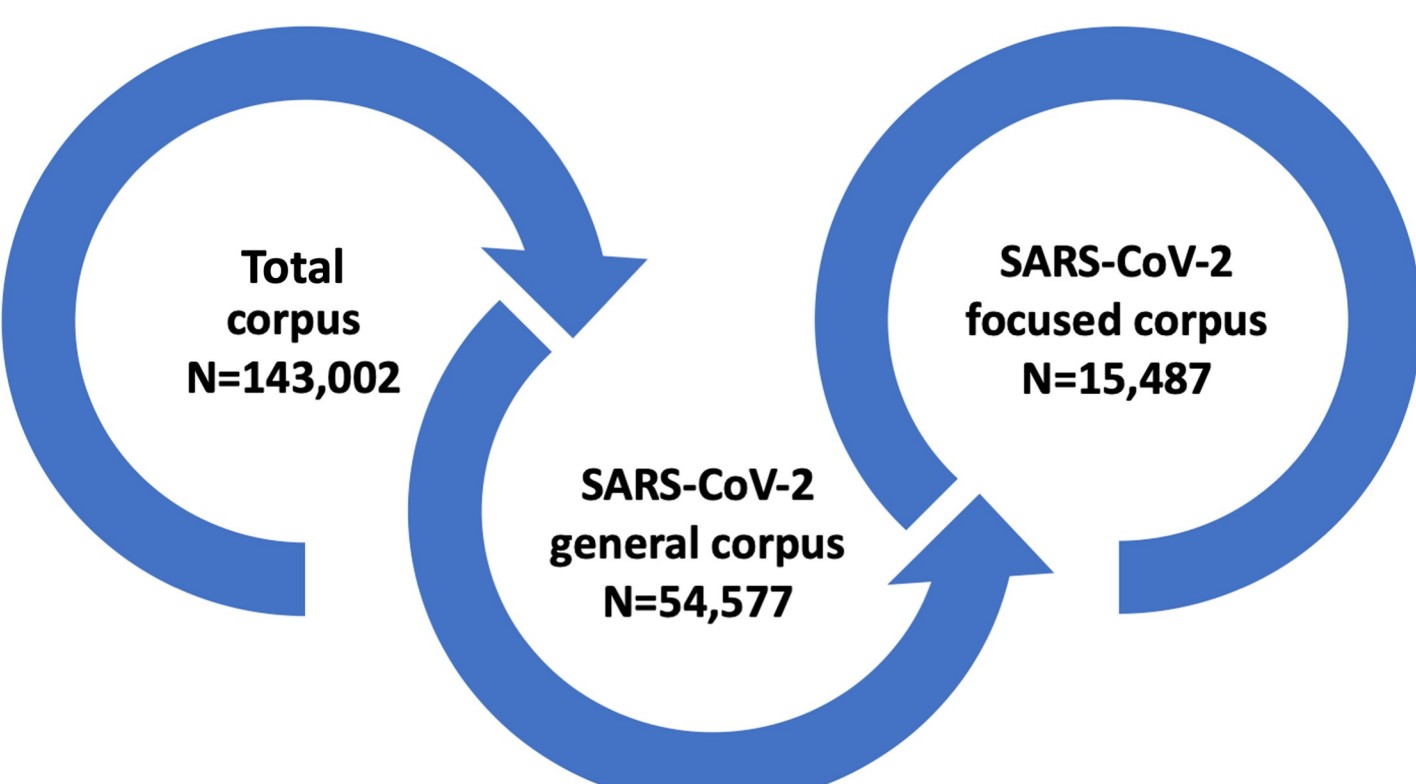

**Fig 1. The process for the composition of the three nested corpora considered within the present study.**

SARS-CoV-2 general corpus was constructed by selecting all the articles published in the time-span between January 1, 2020 and June 15, 2020, which contain at least one of the following terms: [covid, corona virus, OR coronavirus]. The newspaper article harvesting process consists of three main phases: proper article collection, scraping, and de-duplication. Articles had been collected by means of a dedicated media monitoring platform developed within the research initiative "TIPS" (Technoscientific Issues in the Public Sphere, see [38]). Data acquisition relies on online news such as RSS feeds associated with specific newspaper sections obtained through a collector module. Articles with less than 50 characters were excluded because they were mainly short photo gallery or video descriptions.

Corpora pre-processing included tokenization (word unit identification), discarding punctuation, word capitalization (converting to lowercase all capital letters), and filtering out stop-words (functional words such as prepositions, articles, etc.). Even when working through the bag-of-words approach, the main multi-word expressions were analyzed by tokenizing adjacent words into n-grams. In some cases, n-grams were recoded into unigrams or acronyms to monitor whether they appeared among the top words of the topics [39]. This procedure enabled careful detection of certain personalities (politicians, scientists, etc.) and organizations (WHO, ECB, etc.) and assessed their relevance in the topics without ambiguity. Lemmatization was avoided because in large datasets, lemmatizing words can be harmful as it ignores information in the conjugated forms [40].

The "SARS-CoV-2 general corpus" was analyzed using a machine-learning technique, i.e. through both a curated and iterative analysis of topics extracted using LDA. The optimal number of topics was determined by evaluating the results for topic models with different topic numbers. As Di Maggio et al. [29] clarified, model interpretation requires that the data analysts have domain expertise. Accordingly, the following twofold peer-to-peer consensus validation process was adopted: to find labels that fit the content topics well, the list of terms with the highest probabilities of belonging to a topic was carefully reviewed; then, a sample of documents featuring the highest proportions of each topic were read to assure consistency.

By varying the number of topics, several candidate models were run and compared for significant differences, interpretability, and avoiding overlapping between topics [27]. In the first run, 50 topics were extracted from the dataset. All the topic descriptions were then manually scrutinized to select topics pertaining to the Covid-19 pandemic. The selected topics included three major components: (1) explicit reference to issues of healthcare, disease, and illness related to the Covid-19 pandemic; (2) explicit reference to healthcare and biomedical agencies and public policies for managing the Covid-19 pandemic; and (3) explicit reference to biomedical research and medical technologies to address the Covid-19 pandemic. Topics where SARS-CoV-2-related content played a marginal role, even if article text contained the keywords used for the initial query, were excluded (they were mainly connected to other issues such as sport or economic and financial news). Hence, the dataset was refined to only include articles for which one of the selected topics was most relevant (in terms of topic proportion). In this way, through the first LDA run, the number of articles constituting the dataset was reduced from 58,646 to 54,477. This more specific dataset was analyzed through a second LDA run. In this second run, the number of topics for extraction was set to 40 to obtain more specific topics (i.e., boosting their sensitivity to grasp particular and well-bounded issues relevant for addressing the research questions). The most pertinent topics were detected following the same approach adopted in the first run and using the same three inclusion criteria. Subsequently, with the removal of three irrelevant topics (concerning local news or articles related to art and literature), the final number of topics (i.e., 37) was determined by a theoretically motivated choice and obtained through a data-driven approach: in this way, the "SARS-CoV-2 general corpus" was constructed.

Subsequently, a qualitative investigation performed within the "SARS-CoV-2 general corpus" revealed three main thematic domains: politics, science, and medicine (see Section 3.2). The topics included in the three thematic domains were judged by selecting the most pertinent and removing those that were not coherent. Then, the "SARS-CoV-2 focused corpus" was built consisting of 13 topics and 15,487 articles related to the three thematic domains mentioned above.

## Data analysis

As described in the previous section, LDA permitted to extract 37 topics within the "SARS-CoV-2 general corpus" and, consequently, to identify three main thematic domains which compose the "SARS-CoV-2 focused corpus" selecting the articles most closely related to the 13 topics which make explicit reference to science, medicine or politics. Regarding the "focused corpus", the 100 top words (in terms of the probability of being generated from a given topic) for each of the 13 topics were analyzed. After excluding irrelevant terms, the analysis focused on 913 terms. The top words were then distributed into the three categories of organizations, people, and roles, and each of them was related to the following domains of expertise: medical, political, scientific and technical. Overall, 82 terms were classified as entities through these two dimensions (i.e., categories and domains of expertise). Finally, a frequency analysis of this subset was performed in the "focused corpus".

In addition, all three corpora were analyzed to assess the relevance of scientific content within the selected articles by means of a classifier specifically developed for this purpose.

Looking at science as a social activity represented in the media, this study considers "scientifically relevant" an article in which at least two of the following features are mentioned: a scientist; a scientific journal; a research center/laboratory; a scientific discipline (excluding humanities and the social sciences); a generic reference to research processes and/or technological innovations; a discovery, an innovation, a scientific instrument or a medical apparatus. On this basis it is possible to develop a classifier able to decide whether an article can be considered "scientifically relevant" or not [41]. A ground-truth sample was built, manually selecting articles regarded as "scientifically relevant", i.e. featuring at least two of the above-described features (n = 1,167) and articles without those features (n = 2,647).

An initial set of candidate classification algorithms were chosen (a comprehensive overview of the followed approach is provided within the—S3 section in S1 File. See also [38]): Random Forest, Naïve Bayes, Nearest Neighbor, Multinomial Naïve Bayes (MNB), Linear Stochastic Gradient Descent (LSGD), Dual Coordinate Descent method for Logistic Regression (DCD-LR), and Support Vector Machine (Least Squares Support Vector Machine–LS-SVM, and divide-and-conquer solver for kernel SVMs–DC-SVM). We tested these different machine-learning (ML) techniques in terms of $F_1$-score, recall (to minimize the number of false positives), and error rate, through a five-fold hyperparameters cross-validation on a sample of 3,814 articles in Italian that were appropriately labelled. The assessment of the five-fold cross-validation was done on a training set including 80% of the documents randomly selected in the sample (3,051 out of 3,814 documents). Once the optimal values of the hyperparameters for the models were found, their generalizability was compared through the test set, obtained from the remaining 763 articles (20%) of the sample. The best classifier (error rate: 5.70) was obtained by combining and weighing the predictions of two previously selected classifiers–DCD-LR and LSGD–via a stacking ensemble method [42]. This classifier enabled the authors to discriminate between articles with and without relevant scientific content. An "index of salience" (the ratio of articles with scientifically relevant content to the total number of articles in the collection) was calculated to identify scientific salience within the corpora and its trend

across time (see Sections "Italian press in the pandemic: Coverage, scientific salience, and emerging issues" and "Performing public health, institutionalizing science under political jurisdiction").

## Results and discussion

### Italian press in the pandemic: Coverage, scientific salience, and emerging issues

The longitudinal analysis covers the first 5.5 months of the SARS-CoV-2 crisis (January 1, 2020–June 15, 2020) from the alarm the WHO sounded about a new pneumonia outbreak to Italy's almost complete reopening after the lockdown. Fifteen time slots were identified by scrutinizing the key events that punctuated this time range, enabling the authors to better analyze the media coverage of the pandemic following the breaking events that resonated in the media rather than merely focusing on a chronological time flow (see S2 Table in S1 File). For the sake of this paper, it is worth giving a brief overview of the key events in Italy. The country was hit hard by SARS-CoV-2 in late February 2020, almost two months after the WHO's early alarm. While the first registered outbreak concerned people entering Italy from abroad [34, 35], two infection clusters were reported in the Lombardy and Veneto regions, paving the way for the country-wide lockdown declaration (Prime Ministerial Decree on March 9, 2020). After a further and more stringent law decree (Prime Ministerial Decree on March 22, 2020, known as the "Close Italy Decree"), the Government postponed lifting mobility restrictions and reopening commercial activities (April 26, 2020), extending the lockdown for another two months. Italian public health institutions and experts from different disciplines (e.g., emergency medicine, pediatrics and pneumology) supported both the lockdown and the reopening led by the Italian Government under Prime Minister Giuseppe Conte. The experts were gathered into a "Technical–Scientific Committee" (TSC), an ad hoc taskforce charged with offering advice and evidence-based guidelines for managing the SARS-CoV-2 crisis. The media provided massive coverage throughout the aforementioned timespan, with especially heavy coverage across March and April 2020, as reported in [Fig 2].

As expected, SARS-CoV-2 issues were highly covered by newspapers, especially during the two-month lockdown (from March 8, 2020 to May 4, 2020): the main peak is visible exactly from the beginning of the lockdown; its decrease begins in early April. During the lockdown, the query yielded more than 31,000 articles, that is 59% of total of articles published by the eight newspapers during that period. Regarding the relevance of scientific content within the pandemic-based news-making (see RQ1), the "SARS-CoV-2 general corpus" scientific salience was 8.23, whereas it was only 6.22 considering all the articles published. Hence, science, as expected, played a relevant role within the media discourse about the pandemic. Monitoring the evolution of the salience index revealed that the media discourses were grounded in science when the crisis was mainly an *extra moenia* issue [43]—that is, far from being perceived as a direct health threat to Italy. Indeed, as shown in Fig 3, scientific salience was very high in the pre-lockdown phase. Hence, at the very beginning of the time span under analysis, SARS-CoV-2 media accounts were dominated by the worrisome situation in Wuhan, China: data about contagions and the death toll echoed the quest for a scientific explanation of the origin of the unknown virus, its possible threats, and its related risks. Scientific salience subsequently decreased; media narratives about the emergency turned their attention mainly to the social and economic consequences of the virus in the EU and in Italy.

A further relevant element was that media discourse around the pandemic became less engaged with science in the two periods immediately before the lockdown (February 23, 2020–March 1, 2020; March 2, 2020–March 8, 2020), when attention was mainly focused on the

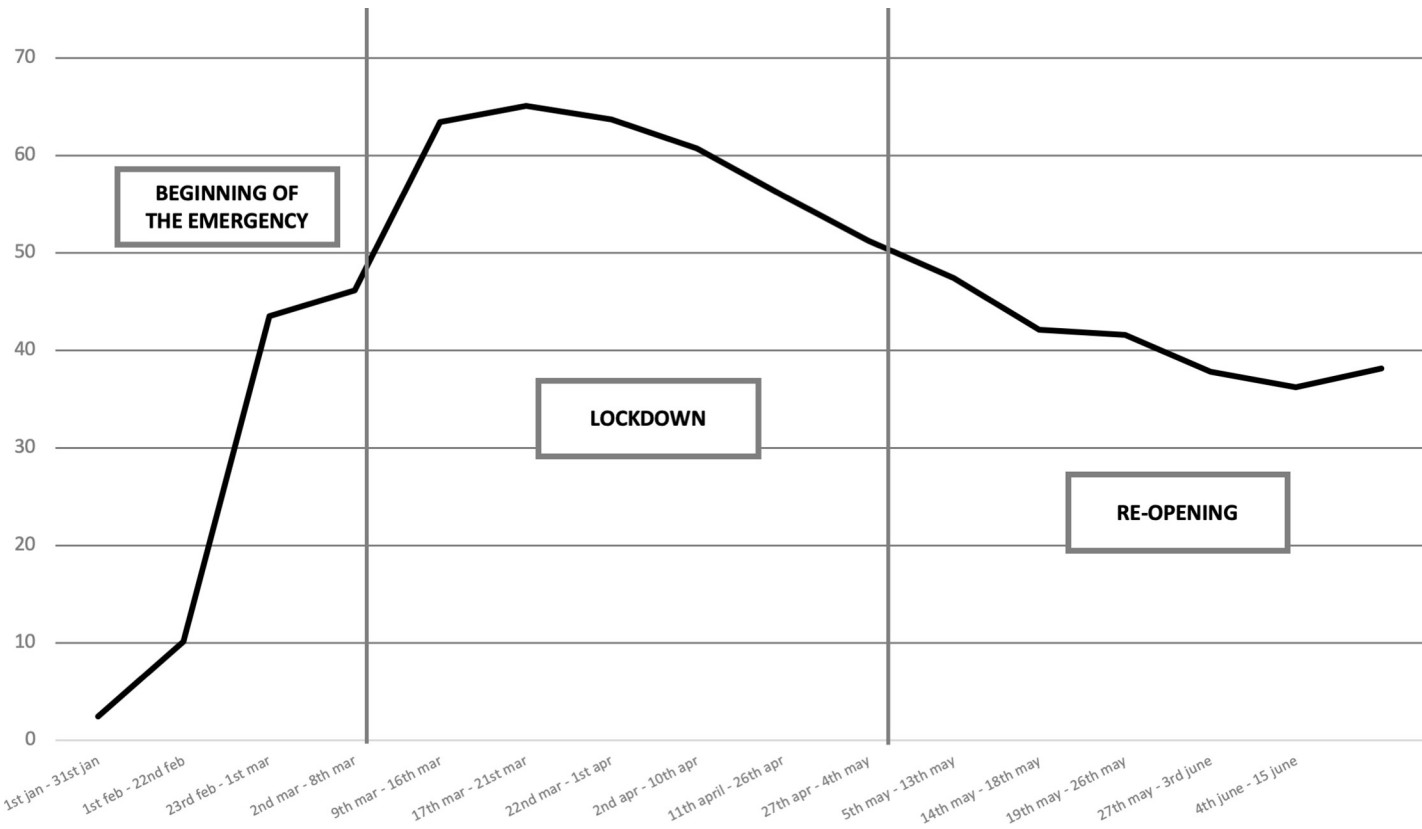

**Fig 2. Distribution across time of the percentage of "SARS-CoV-2 general corpus" articles (N = 54,477) within the "total corpus" the eight Italian daily newspapers published (n = 143,002).**

anti-pandemic regulations and restrictions the Government would be imposing. In contrast, there were two other scientific salience peaks from April 27–May 4 and May 27–June 3. In the first case, the increasing scientific salience was caused by several issues, including uncertainties about open questions linked to post-lockdown actions (the so-called "Phase 2"), pollution as a risk factor for infection, and revamping the debate about the origins of the virus: Was it a zoonosis? Was the virus manufactured in a Chinese biotech laboratory? Beside these questions, the issue of reinfection among people who had already recovered from SARS-Cov-2 emerged. The second peak was related to the controversy about the supposed "weakening" of the virus, expectations regarding a vaccine as well as the actual efficacy of hydroxychloroquine as a treatment.

Hence, in a global context highly affected by scientific uncertainty and controversies, managing the pandemic has been ruled by a complex mix of centralized political decision-making further supported by scientific advice. This connection between science and politics is complexly represented through newspapers. Indeed, as the next sections highlight, while Italian mainstream newspapers portrayed a wider array of discourses brought into the public narrative of the pandemic through different categories of actors, politics played a leading role within the Italian pandemic landscape.

## Framing Sars-Cov-2 as a social, political, and economic virus

Analyzing topic modelling outputs revealed three main thematic domains, according to which the SARS-CoV-2-related media narratives are organized (Fig 4). Firstly, a major political

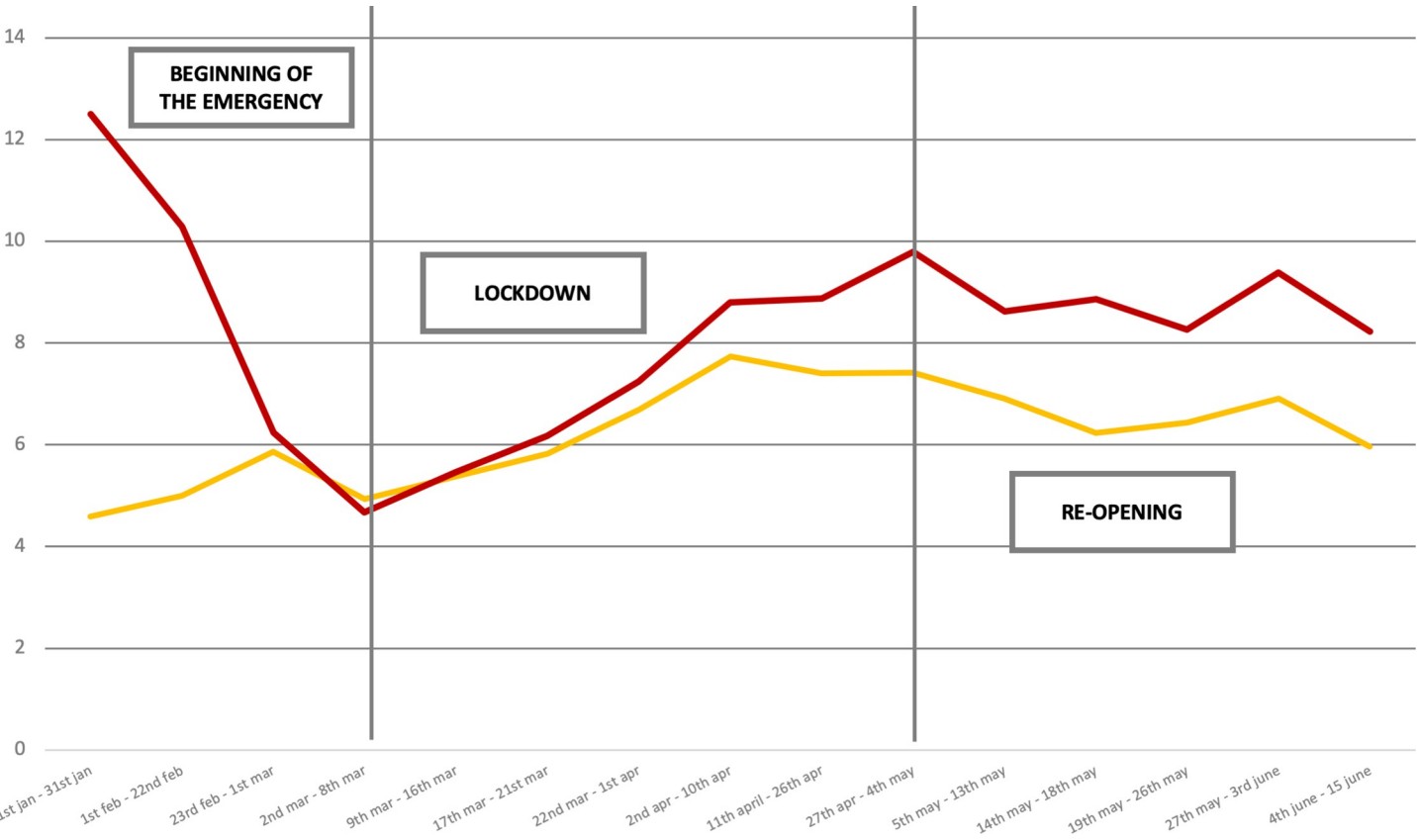

**Fig 3. Scientific salience of the "SARS-CoV-2 general corpus" (in red; μ = 8.23) and of all the articles the eight Italian daily newspapers published (in yellow; μ = 6.22).**

domain composed of seven topics, a scientific domain comprising four topics, and a medical domain comprising two topics were identified. This initial evidence immediately revealed a dominant representation of the pandemic in Italy as a matter of political decision-making.

The number of topics clustered under the politics label reveals the absolute relevance of the political domain in configuring the pandemic media narrative; this is performed through mobilizing actors, stakeholders, institutions, and regulatory tools that primarily pertain to political decision-making. The newspapers here analyzed mainly discussed the pandemic in terms of economic (topic no. 14), social (topic no. 39), and spatial (topic no. 24) relationships; they thus addressed it as a matter of pervasive (self) surveillance practices (topic no. 3). Although the media narratives located the pandemic within the context of individual habits and behaviors—by drawing attention to potential dangerous social practices enabling virus spread (topic no. 35 and no. 39)—it was the "nation" and the common "social organism" that were at risk of becoming ill. Scrutinizing the weekly distribution of articles further confirmed the dominance of the political domain above the others. Indeed, the relevance of topics mainly devoted to scientific research about SARS-CoV-2 (topic no. 37 and topic no. 0), its origins, clinical development, and epidemiological profile decreased over time with the rise of other key topics (e.g., topic no. 3 on lockdown implementation and topic no. 9 on Italian parliamentary politics). Moreover, topics marked as relevant to medicine and science that kept their level constant or ascending over time were mainly connected to the general evolution of the contagion (topic no. 8) and the "push" for a vaccine (topic no. 18). Similarly, topics within the

| Thematic Domain | Topic Number (score) | Topic Label | Top five keywords | Trend across periods |
|---|---|---|---|---|
| Politics | N. 24 (0,221702) | A matter of lockdown: regulations, privacy and freedom in pandemic times | Emergency; Activity; Law; Measures; Health | |
| | N. 3 (0,069230) | Performing lockdown in Italy | Regional; Campania; G. Gallera; Contagion; Piedmont | |
| | N. 9 (0,066378) | Italian parliamentary politics in pandemic times | G. Conte; M. Salvini; Italy; Premier; Emergency | |
| | N. 14 (0,056342) | Pandemic welfare in Italy to face the economic crisis | Bonus; Emergency; INPS; Fund; Families | |
| | N. 35 (0,055594) | Entering Lockdown: what is allowed, how and what is not | Activities; Measures; Decree; Displacements; March | |
| | N. 25 (0,047251) | EU politics and recovery fund in pandemic times | Countries; Italy; Europe; Crisis; European | |
| | N. 39 (0,041969) | Contact tracing to slow the spread of COVID-19 | Data; Contact; Can; Use; App Immuni | |
| Science | N. 37 (0,082433) | Datifying coronavirus spreading in Italy | Cases; Date; Respect; Number; New | |
| | N. 0 (0,077190) | Humans-Animal relationships in pandemic times | Dogs; Species; Cerutti; Fulvio; Pen | |
| | N. 8 (0,076511) | Exploring Covid-19: clinical and epidemiological features | Epidemic; Cases; China; Study; Disease | |
| | N. 18 (0,036697) | The shaping of the bio-clinical gaze over Covid-19 | Vaccine; Test; Drug; Antibodies; Medications | |
| Medicine | N. 5 (0,075409) | Managing public health and hospitals in pandemic times | Hospital; Patients; Doctors; Hospitals; Places | |
| | N. 32 (0,041413) | PPE: production, uses and availability | Masks; Company; Production; Products; EURO | |

**Fig 4. Topics clustered per thematic domain; score (posterior document-topic density), keywords, and trends across the 15 periods were considered.**

medicine domain primarily concerned the tools and equipment needed to manage the emergency, such as personal protective equipment ([PPE], topic no. 32) and great pressure on hospitals created by the volume of patients admitted to intensive care units (topic no. 5). These aspects primarily elicited political responses in the form of a "state of emergency" (topic no. 3, no. 24, and no. 25) as a way to protect national public health and safety and to manage potential political (topic no. 5 and topic no. 9) and economic shocks (topic no. 14) as well as social disintegration (topic no. 5 and topic no. 24). Indeed, the different instances of the emergent SARS-CoV-2 crisis were mainly located in the field of (trans-)national governmental bodies (topic no. 5 and topic no. 25) and parliamentary negotiations between Italian political parties (topic no. 9). Moreover, media narratives mobilized political jurisdiction as crucial to making sense of the illness (first person perspective) related to SARS-CoV-2, where individuals had to act in strict compliance with the emergency laws limiting productive activities and individual mobility (topic no. 3 and no. 24) for the sake of the "nation organism" as a whole. Thus, by evoking a pervasive political jurisdiction, media narratives framed the individual as a subject that was "responsible" and compliant with the pandemic regulatory framework not so much for her or himself but primarily for the social and political community of reference (e.g., the nation, the neighborhood, the kindred, her or his own relational circles). Accordingly, the media shaped a "bio-political community" based on the shared susceptibility of contracting and spreading the virus. In other words, the media portray a specific idea of national community. Rather than being merely geopolitical, they define Italy's common identity by the biological risk of contracting the SARS-CoV-2 infection. Along the same lines, preserving a "healthy

social body" became a central topic in the political agenda, where politicians and political institutions were framed as major actors in charge of providing individuals and families with guidelines and useful resources (both informational and material) to mitigate the consequences of the pandemic in everyday life (see topic no. 14 and no. 25).

Considering the relationship between the three clustered domains, Italian mainstream daily press located science and scientific endeavors (see the second thematic domain in Fig 4) in an ancillary position compared to politics and public governance issues. More precisely, the media addressed science mainly as an organic, well-bounded institution devoted to offering evidence-based insights for legitimizing political decision-making. Further, it did so without ever taking a leading position in reassuring the public or in performing a moral suasion for obtaining citizens' compliance with the measures against the spread of SARS-CoV-2. Science-driven processes were generally circumscribed to the complex and uncertain work of understanding the biological and clinical identity of the concerned virus (see topic no. 8 and topic no. 18) putting the health and wellbeing of the social body at risk. Producing data and collecting clinical evidence were the main duties scientific institutions performed (see topic no. 8 and topic no. 37). Accordingly, the main scientific outputs served as anchor points to extend emergency regulatory political agency. It is worth noting that even if the SARS-CoV-2 had been rapidly genotyped within a few weeks [44] of the first WHO claim about the global spread of the virus, this would mainly have occurred to disavow the hypothesis that it was created in a laboratory. In this scenario, the main health strategy enacted and narrated by the media concerned travel restrictions, quarantine, and the policing of space—that is, measures that political bodies and governmental institutions have been in charge of since the plague of the fourteenth century.

Concerning the medical domain, the narratives were highly related on the one hand to maintaining and optimizing the hospitals as well as care services and, on the other, assessing and managing the risk of contagion from Sars-Cov-2. Special attention was devoted to the production and timely procurement of PPE (see topic no. 5 keywords and topic no. 32); this was a critical issue in the early pandemic stages in Italy. A second focus connected to the medical domain regarded reorganizing healthcare spaces in a pandemic context, and consequently, the debate over administering ordinary care in hospital settings that hosted SARS-CoV-2 patients. In this light, the media was not debating clinical practices and medical knowledge in itself, but redefining health institutions' functioning in accordance with governmental policies discouraging viral spread.

Overall, what emerged was a peculiar style of narrating different actors within their own roles in the pandemic narratives. Accordingly, the next section investigates the relevance of the organizations, people, and roles at play in the pandemic landscape intersected with the medical, political, scientific, and technical domains of expertise.

## Performing public health, institutionalizing science under political jurisdiction

Regarding the types of actors, institutions, and expertise that dominated the SARS-CoV-2 media narratives (RQ2), the "SARS-CoV-2 focused corpus" reveals a preeminent contribution of articles that were more connected to political matters (53.5%), whereas scientific topics represented a far smaller percentage of the corpus (30.3%), and medicine an even smaller one. (16.2%). Thus, once again, politics played a leading role in the media discourse about SARS-CoV-2.

Although the media narrative of the pandemic was characterized by a higher salience compared to other articles not covering the issue, the media scene remained constantly occupied

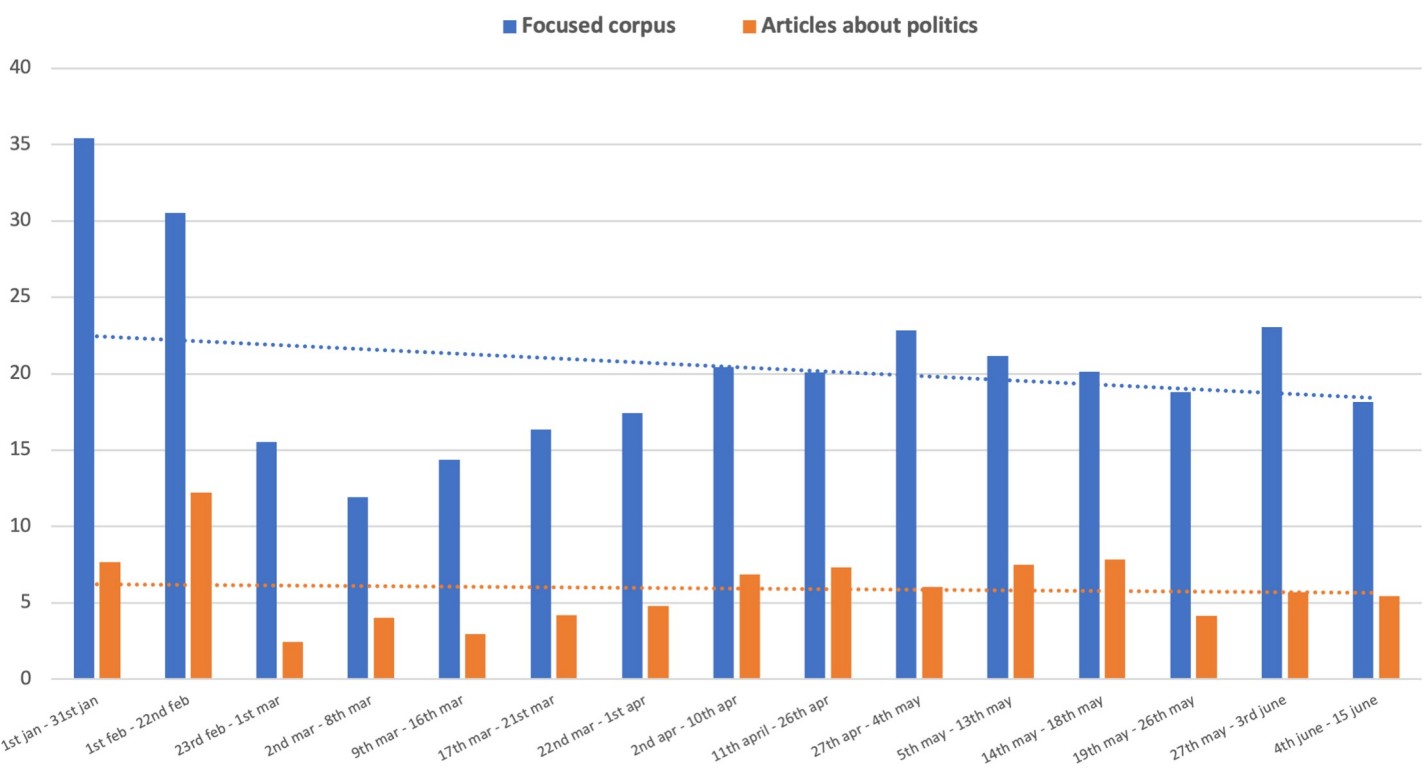

**Fig 5. Trend comparison over time about scientific salience in the "SARS-CoV-2 focused corpus" (μ = 20.41) and in articles with political content (μ = 5.94).**

by politics. Further evidence of this is visible in Fig 3, which plots scientific salience within the "SARS-CoV-2 focused corpus" (blue line). The average value for the period was 20%, twice the value within the "SARS-CoV-2 general corpus" (8.23%). Nevertheless, also in this scenario, characterized by an unusually high scientific salience, media coverage remained constantly hegemonized by politics.

Fig 5 further represents the trend of scientific salience only considering the articles more related to politics (orange line). Scientific content occupies a marginal position when politics is speaking; this position is even smaller in the "SARS-CoV-2 general corpus" (5.94 versus 8.23). Specifically, only 6 out of 100 articles concerning politics also refer to scientific content. A significant exception regards the period from February 1–22, when the pandemic threat was definitively publicly recognized and politicians were facing the problem of "What should we do?". However, right after this initial critical moment of disorientation, science lost its relevance again.

As a consequence, the contribution of science to the media discourse about the pandemic, and more generally, to the collective interpretation of what was going on day by day had two main facets: 1) it provided explanations and suggested practical arrangements/suitable behaviors for laypersons to combat the virus, and 2) it supported political decisions (especially during the initial periods). Indeed, tough and unpopular restrictions, such as limitations to mobility, reducing social interaction, and halting everyday activities (such as work, school, or religious worship) needed to be justified through a politically consistent lens.

Furthermore, while science entered the media discourse almost exclusively as an institution (e.g., the Italian National Institute of Health, the Lazzaro Spallanzani National Institute for Infectious Diseases), politics were mainly enacted through its representatives. As leaders of political parties, ministers, or key members of the Government, they predominantly appeared

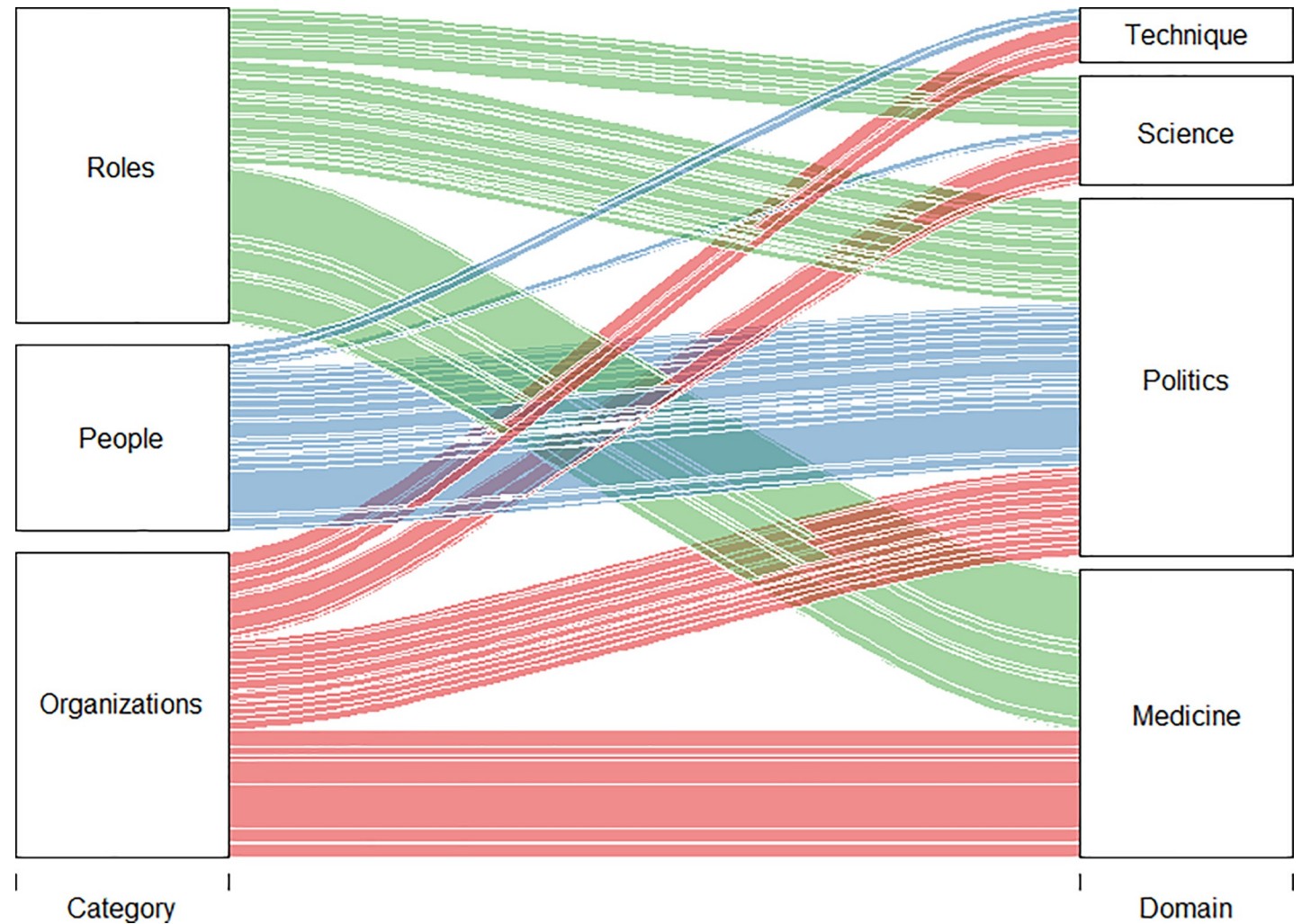

**Fig 6. Named entities (n = 82) distributed by category, domain of expertise, and mutual relationships.**

by name as specific political actors. Prime Minister Giuseppe Conte emerged in a leading position. In some respects, this is not surprising, given the increasing "personalization" of politics [45, 46]. However, it is a specific feature of the pandemic media discourse worth noting, especially when compared to the media salience of scientific content. This aspect emerged by analyzing the words mainly associated with the topics related to the focused corpus.

The personalization of politics and the de-personalized (i.e., institutional) presence of science within the media discourse about SARS-CoV-2 is clearly shown in Fig 6. This illustrates the frequency of the most relevant words related to scientific, political, technical, and medical expertise, in the "SARS-CoV-2 focused corpus". The words were distributed into three categories (roles, people, and institutions) and assigned to four domains (science, politics, medicine, and technical expertise). Under this light, it is possible to observe that the "people" category is fully embedded within political actors, whereas science and technical experts were rarely represented in the media discourse as individuals. On the contrary, science spoke via its institutions (universities, research centers, scientific authorities, etc.) and impersonal roles rather than featuring the names of prominent scientific personalities. Indeed, they were simply referred to based on their profession (e.g., "scientist" or "researcher").

Another element that becomes clear through Fig 6 is that science—notwithstanding its apparently neutral and objective presence in the media discourse—could not avoid being overwhelmed by politics, allowing the Government to exploit science as a source of legitimacy for the measures (of social and economic nature) undertaken. This is further shown by the fact that science appears controversial at times, particularly when its spokespeople enter the media scene. Regarding the pandemic, it was easy to find examples of scientists with contrasting positions in the newspapers, such as on February 20 between the microbiologist Maria Rita Gismondo (which argued: "it is folly to mistake an infection that is a little more serious than the flu for a lethal pandemic") [47] and the virologist Roberto Burioni (which replied: "arguing such a claim is foolish") [48]. Another example is from May 23 between microbiologist Andrea Crisanti and Francesca Russo about the right strategy for mass swab testing to detect asymptomatic infected people to combat viral spread [49].

## Conclusion

The analysis of the mainstream Italian press highlights how the pandemic has been primarily addressed in terms of political regulation. Science lies in the background, while politics battles the virus by exerting its jurisdiction as well as its moral and regulatory authority.

Considering the discourses detected through topic modelling, the political domain is dominant both because it quantitatively overcomes the other domains and because of its decision-making supremacy in determining the conditions, fields, and modalities of public intervention to combat the spread of SARS-CoV-2. Therefore, media narratives across Italian mainstream newspapers do not exclude issues concerning scientific research, epidemiology, and clinical treatment for patients—rather it configures them as a subsidiary body of knowledge to be mobilized for legitimizing the expansion of a political centralized governance of the emergency. Hence, the "cultural authority of science" [33] is questioned, and the public scientific controversies between leading scientists around SARS-CoV-2 issues further boost the centrality of political expertise in managing the health crisis, with a clear success in terms of citizen support as shown by recent research outcomes [50]. Indeed, the media discourse about the pandemic is deeply characterized by politics' "patronization" of science and medicine. Here patronization refers to the pivotal role politics play in defining the frame of reference for connecting biomedical expertise with society, thus providing "its stamp of approval" to public health measures defined by major scientific and medical institutions (e.g. the Istituto Superiore di Sanità [Italian National Institute of Health]). This study allows the authors to argue that patronization unfolds by means of knowledge certification practices unfolded through specific political regulatory tools (e.g. Prime Ministerial Decrees) that allow certain public health measures (e.g. social distancing, curfew, distance learning) to enter the public sphere legitimately. As such, major political actors and political and institutional arrangements neutralize individual scientific actors and the expert knowledge upon which public health measures are based. Therefore such measures, by means of the intermediation of specific political regulatory tools, are brought back to the scientific institution itself, which is depicted as a neutral and objective space for informing political decision making over health crises. In this way, politics can ascribe its pandemic governance action to an alleged expert homogeneity and monovocality of science as an organic institution, thus eliding the fact that pandemic-related scientific discourses can actually evolve into multiple fields comprising a range of different experts and diverging positions over the same debated topics.

In this regard, as strongly demonstrated by the analysis carried out over the people, organizations, and institutions mobilized within the newspaper articles, the political domain is dominant, representing the large majority of active and relevant actors. In contrast, other domains

are de-personalized; the scientific, medical, and technical authorities, although not absent, play mainly institutional supporting roles. Lastly, media narratives exalt the personification of politics, reducing science and medicine to institutional roles.

Overall, the mainstream media accounts of SARS-CoV-2 have enabled the shaping of a peculiar form of political jurisdiction that can be labelled in terms of "pastoral power": a kind of knowledge deployed by politicians in cooperation with scientific counsellors (i.e., the Technical and Scientific Committee). This knowledge can be extended to encompass predictive and future-oriented information based upon evidence such as the epidemiological profile of the pandemic, that may indicate risk of future spreading of Sars-CoV-2 or undesirable behaviors like social proximity. The sites of this jurisdiction proliferate in different social, cultural, and economic fields and are irreducible to the mere scientific or medical spheres and actors. This jurisdiction that the media impute to politicians espouses the ethical principles of a "hygienist frame," where citizens are required to take responsibility for their own medical futures as well as those of their families and children. Thus, these "ethical principles" are translated into public policies to manage pandemics that are inescapably normative and directional; they cover the social and the economic relationships within the country.

## Supporting information

**S1 File.**
(DOCX)

## Acknowledgments

This work has been conducted within the research initiative "TIPS" (Technoscientific Issues in the Public Sphere), which developed the research platform used in this article. TIPS is scientifically chaired by Professor Federico Neresini (University of Padova) and hosted by the Research Unit "Padova Science, Technology and Innovation Studies" (University of Padova). We are grateful to Alberto Zanatta for his valuable support in data collection and topic modelling analysis, and to Virginia Zorzi for her precious help in revising the English version of this text.

## Author Contributions

**Conceptualization:** Stefano Crabu, Paolo Giardullo, Federico Neresini.

**Data curation:** Paolo Giardullo, Andrea Sciandra.

**Formal analysis:** Andrea Sciandra.

**Methodology:** Andrea Sciandra, Federico Neresini.

**Supervision:** Stefano Crabu, Federico Neresini.

**Writing – original draft:** Stefano Crabu, Paolo Giardullo, Andrea Sciandra, Federico Neresini.

**Writing – review & editing:** Stefano Crabu.

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
