## [Decision Letter · Decision Letter 0]

2 Mar 2021

PONE-D-20-40039

Politics overwhelms science in the SARS-CoV-2 pandemic: evidence from the whole coverage of the Italian quality newspapers

PLOS ONE

Dear Dr. Crabu,

Thank you for submitting your manuscript to PLOS ONE. After careful consideration, we feel that it has merit but does not fully meet PLOS ONE’s publication criteria as it currently stands. Therefore, we invite you to submit a revised version of the manuscript that addresses the points raised during the review process.

All the reviewers appreciated your work, however they identified a series of issues that need to be addressed before publication. I would invite the authors to consider all the reviewers' suggestions and comments, especially w.r.t. the contextualisation of the work in the literature, clarifications on the methodology used in the paper, and the definitions of science and scientific source.

We look forward to receiving your revised manuscript.

Kind regards,

Fabiana Zollo, Ph.D.

Academic Editor

PLOS ONE

Journal Requirements:

Reviewers' comments:

Reviewer's Responses to Questions

**Comments to the Author**

1. Is the manuscript technically sound, and do the data support the conclusions?

Reviewer #1: Partly

Reviewer #2: Yes

Reviewer #3: Yes

2. Has the statistical analysis been performed appropriately and rigorously? 

Reviewer #1: Yes

Reviewer #2: Yes

Reviewer #3: Yes

3. Have the authors made all data underlying the findings in their manuscript fully available?

Reviewer #1: Yes

Reviewer #2: No

Reviewer #3: Yes

4. Is the manuscript presented in an intelligible fashion and written in standard English?

Reviewer #1: Yes

Reviewer #2: Yes

Reviewer #3: Yes

5. Review Comments to the Author

Reviewer #1: The manuscript offers a quantitave analysis of the media coverage (main Italian newspapers) of politics and science during the first months of COVID-19 outbreak in Italy. The main topics discussed about the two news categories are also investigated as well the distribution of the media attention on different types of actors: role, individual, organization.

As result of the analysis, the authors conclude that the health emergency has been addressed primarily in terms of political regulation and concerns only marginally as a scientific matter. Moreover, the analysis reveals that the personalization in media coverage, only understood as individualization (see [1]), is a phenomenon typical of politics.

I find the paper and the arguments discussed of interest. Nevertheless, I believe that the observation period should be extended for increasing the robustness of the results. Indeed, despite the timespan of about 6 months and despite the scientific nature of the thematic domain, I guess that such period does not represent a suitable ground for investigating the scientific salience of news items about COVID-19 and then for supporting the claim inferred by answering RQ1. Basically, the highlighted findings only confirm that the times of science are very different from those of journalism and not consistent with the early solutions required during health emergencies such as the one in progress [2]. This could be also inferred by reading the breaking events identified for dividing the timewindow into disjoint periods (S2 Tab). To date, there are no approved therapies which are known to be really efficient in fighting the coronavirus desease. Excluding few news emphasing the potential effectiveness of known drugs or experimental therapies, their early rejection by the competent authorities has soon reduced the media attention. Moreover, the scientific community has mainly focused its efforts on the development of vaccines, but the media attention on this topic only began when the first vaccines has been approved and their purchase and administration has began. Last, the authors should stress that not only politics has long been mediated, but also the process of mediatization of politics is now completely accomplished. On the contrary, many important scientists, especially virologists, have achieved notoriety only during the COVID-19 pandemic. This could partially (or totally) justify the claim answering to RQ2 (Fig. 6). Furthermore, this sudden popularity combined with the initial lack of information and data, has led many scientists go on personal interpretations of the virus and its harmfulness, with the resultant spread of news counteract each other (lines 389-398).

I suggest to query the media monitoring platform already used, in order to extend the analysis to at least the last six months (oct 2020 - feb 2021), roughly coinciding with the second wave of pandemic and the real vaccines debate. If the authors decide to follow my suggestions, I stress the importance to pay great attention on the classification of contents about vaccinations as political or scientific.

The recursive use of LDA is a common procedure in topic modeling tasks, so nothing to say about the methodology.

Two typos at line 31 and 45, intruding 'and' and space, respectively.

[1] P. Van Aelst, T. Sheafer, J. Stanyer, The personalization of mediated political communication: A review of concepts, operationalizations and key findings , Journalism 13 (2), pp 203-220, 2012

[2] COVID-19, the public debate on social media. Available at https://agcom-ses.github.io/COVID/social_media.html?lang=en

Reviewer #2: Referee Report

The paper enters into the scientific debate regarding the mediatic effect of the COVID-19. Specifically, the author(s) investigates the interaction between science and political arguments in the media. The work considers the Italian's mainstream newspapers, assuming that pandemics are strictly interconnected with the broader media, cultural, and political landscapes. The source of data employed represents the mainstream media's covering considering the most common political and cultural positions within Italian society. From a methodological perspective, the paper presents the results from a combination of traditional machine learning tools applied to the selected corpora.

Despite the rigorous application of the selected methodological items, the following minor points must be considered before publication:

• From a general point of view, I recommend looking at

o [2] for better positioning the present manuscript in the scientific debate. As it is, the present work addresses crucial aspects, but it has no mention of the relevant literature around the global "infodemic" clearly connected with the topic.

o [1,3] contain detailed steps of corpora pre-processing. The author(s) can briefly describe what is happened to their texts before feeding the ML algorithms (this point is further stressed later).

• A few words and references for supporting the media's division in: progressive, moderate, conservative, neoliberal and Catholic is undoubtedly beneficial. As it is, one can question the subjective interpretation.

• In the Introduction, the author(s) writes, "This is particularly urgent in a context of relative lack of curative and preventive treatments to face COVID-19 disease, where policies and protocols against the spreading of the virus are primarily rooted in lifestyle and behavioural changes, that is social norms and convention, whose plausibility and legitimacy are widely debated by mainstream media." At the moment of this revision, this is not 100% true anymore (see vaccination campaigns). I suggest an edit of statements like these to stay 100% true for the current period and hopefully at the moment of the publication.

• Regarding the RQ1 "Or more specifically, which domain, between the scientific and the political one, is prevalent in media discourses over SARS-CoV-2 pandemic? "I reckon that the intention of the author(s) is to determine the interaction between the scientific and the political debate. To determine the prevalence of one to the other requires the definition of a – or more -measure of such a phenomenon. This aspect would benefit from further clarifications.

• Regarding the data and its processing:

o The creation of the corpus used for the analysis should be in the body of the paper and not in the appendix as well as a clearer description of the procedure to prepare the texts (e.g. what did you do with stop words? Have you used a Bag of words approach? TF-IDF?).

o When the author(s) says: "The three corpora were analyzed using machine learning techniques to obtain a classification according to their content (scientific or non-scientific) and to identify the topics covered by the articles." Which are the 3 corpora? This should be further clarified in the body of the paper, maybe when the data is introduced (the 3 corpora are mentioned in section "MATERIAL AND METHODS" but properly introduced in "DATASET"). In the current version, one can understand that just after having looked at the appendix.

o Furthermore, in the manuscript's body, the claim of using ML techniques as it stands is too vague. Similarly, in section "Data Analysis", the ML methods mention does not help in clarifying the description of the ML techniques used. Details about it are reported in the appendix only. The modelling decisions are too relevant for being left in there.

o The LDA is introduced later in 144 e 145. This is too late in the text; a more careful reading and a subsequent reorganization is necessary.

• Regarding the topics met during the final run of the topic classifier, it would be appreciated a distribution of the recurrent topics meet in the corpora chosen. This can help the reader in mapping.

• In the "Data Analysis" section, it is not clear how the training and the test data set have been identified if the considered articles' label were not known apriori. They might have been manually labelled or something else; It will be beneficial to state it more clearly.

• Regarding the conclusions: the author(s) wrote, "Indeed, the media discourse about the pandemic is deeply characterized by politics' "patronization" of science and medicine." This should be more explicitly connected with the results obtained. The results look promising, but a futher contexutalizaiton will surely help highlight them, maybe with a few examples that could seat in the appendix if the author(s) does not want to have them in the paper. Similar comments apply to the following statement in the manuscript "Lastly, media narratives exalt the personification of politics, reducing science and medicine to institutional roles."

Bibliography

[1] Cinelli, M., Ficcadenti, V., & Riccioni, J. (2019). The interconnectedness of the economic content in the speeches of the US Presidents. Annals of Operations Research, 1-23.

[2] Cinelli, M., Quattrociocchi, W., Galeazzi, A., Valensise, C. M., Brugnoli, E., Schmidt, A. L., ... & Scala, A. (2020). The covid-19 social media infodemic. Scientific Reports, 10(1), 1-10.

[3] Ficcadenti, V., Cerqueti, R., & Ausloos, M. (2019). A joint text mining-rank size investigation of the rhetoric structures of the US Presidents' speeches. Expert Systems with Applications, 123, 127-142.

Reviewer #3: The topic modelling and the use of LDA allows the analysis of a large dataset of articles and offers a very interesting general overview of mainstream media coverage of the first months of the pandemics in Italy. It also identifies the dominant tone of voice of the coverage, that privileged the political discourse over the scientific one.

Nonetheless this study has a major flaw: the disentaglement between science and politics, in such an uncertain frame, is hard to perform using only salience as a marker. Scientific evidence is influenced by politics and ideology, as demonstrated by the case of hydroxychloroquine (see https://www.thelancet.com/journals/lancet/article/PIIS0140-6736(20)32221-2/fulltext and https://www.nature.com/articles/s41562-020-0894-x for the debate on this topic).

The authors should clarify how they defined "science" and how they disentangled science from politics in the data analysis (the selection of a set of keywords is not enough, as it doesn't allow to evaluate the appropriateness of the scientific information. The same set of keywords could easily identify an article supporting pseduscientific views).

Another important point that the paper does not clarify is the source of the scientific evidence when presented by the media. Italian journalists tend to rely more on experts' advices than on researches or peer-reviewed papers. Is the personal opinion of the experts classified as scientific content (and should we consider it scientific or political)? On the role of experts and the epistemic authorithy in the pandemic see https://www.frontiersin.org/articles/10.3389/fpubh.2020.00356/full

6. PLOS authors have the option to publish the peer review history of their article (what does this mean?). If published, this will include your full peer review and any attached files.

Reviewer #1: No

Reviewer #2: No

Reviewer #3: **Yes: **Daniela Ovadia (University of Pavia; Center for Ethics in Science and Journalism - Milan)

---

## [Author Response · Author response to Decision Letter 0]

6 Apr 2021

APRIL 2021

Dear Reviewers and Assigned Academic Editor,

Thank you for the opportunity to revise and resubmit the manuscript “Politics overwhelms science in the Covid-19 pandemic: evidence from the whole coverage of the Italian quality newspapers” to be considered for publication as a research article in PLOS ONE.

The reviewers’ suggestions were highly insightful and enabled us to improve the quality of our manuscript since they identified conceptual and methodological issues that needed clarification.

We believe we have found a solution that addresses all the main issues and concerns raised by the reviewers. Accordingly, we have integrated the required revisions, and we highlighted the changes to our manuscript within the document by using colored text.

Below you can find our point-by-point responses to the major comments and concerns of the reviewers and assigned academic editor. 

♦ As required by rev.#2 and by the assigned academic editor, we have better positioned the manuscript within the current related scientific debates, both by mentioning the emerging contributions about the so-called “infodemic”, and by providing a more thorough clarification of our theoretical frame in addressing media (see section “Introduction”). In this respect, as rightly argued by rev. #1, although it is well recognized that the pace of newsmaking is quite different from that of scientific knowledge-making, for the purposes of our paper it is relevant that: ii) on one hand, media can be scrutinized as active agents contributing themselves to the development of the ways of managing the pandemic (i.e. media as performative agent; on this point see among others Cartwright 1998; Clarke 2009; Altheide, 2013); ii) on the other hand, media represent a valuable source of empirical data precisely for studying those processes. Following this theoretical frame, we clarified how in our study media are understood both as discursive arenas engaged in shaping public responses to the pandemic, and as a data source for analyzing how political institutions manage relations with scientific regulatory bodies and scientific communities for the sake of public health. Hence, in this perspective the concern is not so much to describe the ways in which media depict science and pandemic public policies; rather we focus our attention on how the concerned media can provide evidence about the reconfiguration of the nexus of science and politics during the early phases of the Covid-19 pandemic.

♦ As required by rev.#1 and by the assigned academic editor, we provided a fully methodological rationale for our decision to focus exclusively on the so-called first pandemic wave (see lines 106-117). It is worth mentioning that we decided to focus on this timespan (January-June 2020) characterized by a relative lack of curative and preventive treatments for the Covid-19 disease, in which public health responses against the spread of the Sars-Cov-2 were primarily rooted in lifestyle and behavioral changes, that is social norms and conventions, whose plausibility and legitimacy can be widely debated by mainstream media. Furthermore, it is crucial to clarify that the later timespan of the subsequent pandemic waves clearly shows a media landscape in which the nexus between science and politics is primarily centered around and dominated by the vaccination campaign debate. Hence, this aspect strongly hampers a thorough understanding of how the re-articulation of the scientific and political poles (in which the subsequent vaccination campaigns are rooted) actually occurred, and which effects it produced on the public governance of the health crisis;

♦ As required by rev.#2, in the section “Conclusion” we located more explicitly within the findings the notion of “politics’ patronization” in relation to the “personalization of politics” in media accounts by further discussing how patronization refers to the key role of politics in defining the frame of reference for interfacing biomedical expertise and society, thus providing “its stamp of approval” to public health measures defined by major scientific and medical institutions. The findings allow us to argue that patronization unfolds by means of knowledge certification practices developed through specific political and institutional arrangements that allow certain public health measures to enter legitimately in the public sphere. As such, major political actors and political institutional arrangements neutralize individual scientific actors and the expert knowledge on which concerned public health measures are grounded. Hence such measures, by means of the intermediation of specific political regulatory tools, are brought back to the scientific institution itself, depicted as a neutral space for informing political decision-making over health crises. In this way, politics can impute its pandemic governance action to an alleged expert homogeneity and monovocality of science as an organic institution, thus eliding the fact that pandemic scientific discourses can actually evolve into pluricentric fields comprising a range of different experts and diverging positions over the same debated topic.

♦ As required by rev.#3 and by the assigned academic editor, we fully describe how we address what can be defined as the demarcation problem between science and other thematic domains (see sections “Introduction” and “Data Analysis”), by clarifying both how we selected the articles in order to establish the corpora we analyzed by the means of topic-modelling techniques and how we build our classifier to discriminate the content of newspaper articles in terms of their scientific relevance. In doing so, we emphasize that we addressed the “demarcation problem” in an empirically-oriented manner. In this regard, even if our approach does not theoretically address the demarcation of science from other social activities (which is an issue quite far from the purposes of this study; see Gieryn, 1983), the article suggests that this problem is to be addressed in terms of a gradient of intersection of science with other fields rather than providing a clear distinction between well-bounded domains. 

♦ As required by rev.#2, we included references to support the classification of the media as progressive, moderate, conservative, neoliberal and Catholic, which is also a well-recognized set of categorizations used by political sociologists and political scientists (see line 209).

♦ As required by rev.#2, we described the pre-processing phase in the “Datasets” section, in particular by including details regarding the corpus creation, text preparation, stopwords, and the bag-of-words approach. 

♦ As required by rev.#2, we clarified the issue of the three corpora as follows: in the “Materials and Methods” section we only mentioned the existence of several corpora, and we referred to the “Datasets” section for further details. In this latter section we outlined the steps for constructing each of the three corpora.

♦ As required by rev.#2, we added details about the Machine Learning (ML) methods we used. In the “Data Analysis” section we included initial candidate classification algorithms, the cross-validation procedure, the creation of training and test sets, and the manual labeling of those articles in order to extend the classification to our corpora through Supervised ML.

♦ As required by rev.#2, we pointed out in the “Introduction” that our topic modeling is based on the Latent Dirichlet Allocation (two sections earlier than in the original manuscript).

♦ As required by the assigned academic editor, and following the Plos One Policies, we made the “Sars-Cov-2 general corpus” data publicly available. Since the copy-rights of the article full texts are owned by several third-party organizations (i.e., the publishers of the eight newspapers), we provided all information necessary for interested researchers to gain access to the data: date, source (newspaper), and URL of each of the 58,646 articles belonging to the “SARS-CoV-2 general corpus.” This dataset is adequate to replicate our study findings, and it has been deposited in a public repository, Zenodo, under the following doi: 10.5281/zenodo.4624096 (Data from: Politics overwhelms science in the Covid-19 pandemic: evidence from the whole coverage of the Italian quality newspapers).

♦ The title has been modified by replacing “Sars-Cov-2 pandemic” with “Covid-19 pandemic”, which is actually more accurate for referring to the current global disease outbreak. 

********

Once again, our grateful thanks for your care and attention. We hope that we have made good use of your efforts. 

The authors

---

## [Decision Letter · Decision Letter 1]

10 May 2021

Politics overwhelms science in the Covid-19 pandemic: evidence from the whole coverage of the Italian quality newspapers.

PONE-D-20-40039R1

Dear Dr. Crabu,

We’re pleased to inform you that your manuscript has been judged scientifically suitable for publication and will be formally accepted for publication once it meets all outstanding technical requirements.

Kind regards,

Fabiana Zollo, Ph.D.

Academic Editor

PLOS ONE

Additional Editor Comments (optional):

Reviewers' comments:

Reviewer's Responses to Questions

**Comments to the Author**

1. If the authors have adequately addressed your comments raised in a previous round of review and you feel that this manuscript is now acceptable for publication, you may indicate that here to bypass the “Comments to the Author” section, enter your conflict of interest statement in the “Confidential to Editor” section, and submit your "Accept" recommendation.

Reviewer #1: All comments have been addressed

Reviewer #2: All comments have been addressed

2. Is the manuscript technically sound, and do the data support the conclusions?

Reviewer #1: Partly

Reviewer #2: Yes

3. Has the statistical analysis been performed appropriately and rigorously? 

Reviewer #1: Yes

Reviewer #2: Yes

4. Have the authors made all data underlying the findings in their manuscript fully available?

Reviewer #1: Yes

Reviewer #2: Yes

5. Is the manuscript presented in an intelligible fashion and written in standard English?

Reviewer #1: Yes

Reviewer #2: Yes

6. Review Comments to the Author

Reviewer #1: (No Response)

Reviewer #2: The comments have been addressed and I am convinced about the improvements made in the present version of the manuscript.

7. PLOS authors have the option to publish the peer review history of their article (what does this mean?). If published, this will include your full peer review and any attached files.

Reviewer #1: No

Reviewer #2: No

---

## [Editor Report · Acceptance letter]

12 May 2021

PONE-D-20-40039R1 

Politics overwhelms science in the Covid-19 pandemic: evidence from the whole coverage of the Italian quality newspapers. 

Dear Dr. Crabu:

I'm pleased to inform you that your manuscript has been deemed suitable for publication in PLOS ONE. Congratulations! Your manuscript is now with our production department. 

Kind regards, 

on behalf of

Dr. Fabiana Zollo 

Academic Editor

PLOS ONE